# Genomic Surveillance of *Listeria monocytogenes* in Taiwan, 2014 to 2019

Yu-Huan Tsai,[a] Alexandra Moura,[b,c] Zi-Qi Gu,[a] Jui-Hsien Chang,[d] Ying-Shu Liao,[d] Ru-Hsiou Teng,[d] Kuo-Yao Tseng,[a] Dai-Ling Chang,[a] Wei-Ren Liu,[a] Yu-Tsung Huang,[e,f] Alexandre Leclercq,[b,c] Hsiu-Jung Lo,[g,h,i] Marc Lecuit,[b,c,j] Chien-Shun Chiou[d]

[a]Laboratory of Host-Microbe Interactions and Cell Dynamics, Institute of Microbiology and Immunology, National Yang Ming Chiao Tung University, Taipei, Taiwan
[b]Biology of Infection Unit, Institut Pasteur, Université Paris Cité, Inserm U1117, Paris, France
[c]Institut Pasteur, National Reference Centre and WHO Collaborating Centre *Listeria*, Paris, France
[d]Center for Diagnostics and Vaccine Development, Centers for Disease Control, Taichung, Taiwan
[e]Departments of Laboratory Medicine and Internal Medicine, National Taiwan University Hospital, National Taiwan University College of Medicine, Taipei, Taiwan
[f]Graduate Institute of Clinical Laboratory Sciences and Medical Biotechnology, National Taiwan University, Taipei, Taiwan
[g]National Institute of Infectious Disease and Vaccinology, National Health Research Institutes, Miaoli, Taiwan
[h]School of Dentistry, China Medical University, Taichung, Taiwan
[i]Department of Biological Science and Technology, National Yang Ming Chiao Tung University, Hsinchu, Taiwan
[j]Division of Infectious Diseases and Tropical Medicine, Institut Imagine, APHP, Necker-Enfants Malades University Hospital, Paris, France

Yu-Huan Tsai and Alexandra Moura contributed equally. Author order was determined by mutual agreement.
Marc Lecuit and Chien-Shun Chiou share senior authorship. Author order was determined by mutual agreement.

**ABSTRACT** *Listeria monocytogenes* is a life-threatening foodborne pathogen. Here, we report the genomic characterization of a nationwide dataset of 411 clinical and 82 food isolates collected in Taiwan between 2014 and 2019. The observed incidence of listeriosis increased from 0.83 to 7 cases per million population upon implementation of mandatory notification in 2018. Pregnancy-associated cases accounted for 2.8% of human listeriosis and all-cause 7-day mortality was of 11.9% in nonmaternal-neonatal listeriosis. *L. monocytogenes* was isolated from 90% of raw pork and 34% of chicken products collected in supermarkets. Sublineages SL87, SL5, and SL378 accounted for the majority (65%) of clinical cases. SL87 and SL378 were also predominant (57%) in food products. Five cgMLST clusters accounted for 57% clinical cases, suggesting unnoticed outbreaks spanning up to 6 years. Mandatory notification allowed identifying the magnitude of listeriosis in Taiwan. Continuous real-time genomic surveillance will allow reducing contaminating sources and disease burden.

**IMPORTANCE** Understanding the phylogenetic relationship between clinical and food isolates is important to identify the transmission routes of foodborne diseases. Here, we performed a nationwide study between 2014 and 2019 of both clinical and food *Listeria monocytogenes* isolates and sequenced their genomes. We show a 9-fold increase in listeriosis reporting upon implementation of mandatory notification. We found that sublineages SL87 and SL378 predominated among both clinical (50%) and food (57%) isolates, and identified five cgMLST clusters accounting for 57% of clinical cases, suggestive of potential protracted sources of contamination over up to 6 years in Taiwan. These findings highlight that mandatory declaration is critical in identifying the burden of listeriosis, and the importance of genome sequencing for a detailed characterization of the pathogenic *L. monocytogenes* genotypes circulating in Asia.

**KEYWORDS** core genome multilocus sequence typing, *Listeria monocytogenes*, listeriosis, whole-genome sequencing

Address correspondence to Chien-Shun Chiou, nipmcsc@cdc.gov.tw.

The authors declare no conflict of interest.

10.1128/spectrum.01825-22   **1**

Listeriosis is a foodborne infection caused by the Gram-positive bacterial pathogen *Listeria monocytogenes*. Human listeriosis is a severe infection characterized by bacteremia, meningoencephalitis, and maternal-neonatal infection (1, 2). While most human foodborne infections are associated with a high incidence but by a relative low morbidity and mortality, listeriosis has a low incidence varying between 0.1 and 12 per million people, but a high mortality rate (15% to 21%), even when active antimicrobial therapy is prescribed (3–5).

The population structure of *L. monocytogenes* is highly clonal, composed of four phylogenetic lineages, 15 serotypes that can be grouped into six PCR serogroups (6, 7), and hundreds of clonal complexes (CCs) and sublineages (SLs) as defined by multilocus sequence typing (MLST) and core genome MLST (cgMLST), respectively (8). Multiple listeriosis outbreaks associated with contaminated food products have been reported worldwide (9–13), mainly serogroups IVb (lineage I) and IIa (lineage II). The virulence of *L. monocytogenes* is heterogeneous: some CCs are hypervirulent and more efficient in gut colonization (e.g., CC1, CC2, CC4, and CC6 from serogroup IVb), resulting in high clinical frequency; others (e.g., CC121 and CC9, from serogroups IIa and IIc) are hypovirulent, and express a truncated form of internalin (InlA) (14, 15), a major virulence factor of *L. monocytogenes* (16, 17). The hypovirulent clones exhibit higher tolerance to surface disinfectants and are more prevalent in food isolates, infecting mainly highly immunocompromised individuals (15, 18).

In Asia, the incidence of listeriosis was thought to be low before the early 2000s. However, an increase of listeriosis incidence comparable to that of Western countries has been observed in the last decade (19–23), likely as a consequence of enhanced surveillance. In Taiwan, patient information and isolates of notifiable diseases are collected by the Taiwan Centers for Disease Control. No listeriosis outbreak had been reported prior to mandatory notification and only a few cases have been sporadically identified in medical centers, mostly associated with CC87, CC19, and CC155 (20, 24). While listeriosis has been a notifiable disease in many Western countries since the 1990s, it was only implemented in Taiwan in January 2018, which led to a significant increase in reported listeriosis cases. Here, we provide a comprehensive nationwide overview of listeriosis incidence prior and after mandatory declaration and a detailed genome-based characterization of *L. monocytogenes* genotypes circulating in Taiwan.

## RESULTS

**Epidemiology of listeriosis in Taiwan.** We included 411 cases of human listeriosis in this study. Before nationwide mandatory notification, only 78 listeriosis cases were reported from 2014 to 2017 in 10 cities of Taiwan, and an incidence of 75 cases per million people was estimated based on total patient admissions to a medical center in Taipei (20). Following mandatory notification, 333 human listeriosis cases were reported in Taiwan in 2018 to 2019 and disease information was available from 327 of them. These comprised 278 (85%) bacteremia cases, 32 (9.8%) central nervous system (CNS) infections, nine (2.8%) maternal-neonatal infections, and eight of other infections (peritonitis, $n = 5$, cholecystitis, $n = 2$, pleurisy, $n = 1$) (Fig. 1A; Table 1). The annual incidence was of 7.16 and 6.95 listeriosis cases per million people in 2018 and 2019, respectively (Fig. 1A). Cases were mainly in the main island of Taiwan, with very few cases in the offshore islands (two cases in Penghu and one case in Kinmen) (Fig. 1B).

The median age of the patients was 68.5 years (range = 0 to 96 years) for the bacteremia cases, 65.5 years (range = 20 to 93 years) for the CNS infections, 60 years (range = 41 to 87 years) for the peritonitis and cholecystitis, and 30 years (range = 26 to 38 years) for the maternal infections (Table 1). Diarrhea was noted in 5.71% of the patients and only in bacteremia patients (6.47%) (Table 1). Among the 318 nonmaternal-neonatal listeriosis cases, 166 (52.2%) were female (Table 1). All-cause 7-day mortality was 11.9% in nonperinatal listeriosis (Table 1).

**Prevalence of *L. monocytogenes* in food samples.** *L. monocytogenes* was isolated from raw chicken products (25 of 113, 22.12%) and raw ground pork products (69 of 117, 58.97%) (Table S1). Notably, 90.41% (66/73) of raw ground pork products in the supermarkets were positive for *L. monocytogenes*, whereas only 6.8% (3/44) of those sampled in the open-air markets were positive for *L. monocytogenes*. In the case of chicken products, 33.8% (25/74) of those sold in supermarkets were positive for *L. monocytogenes*, and all the chicken

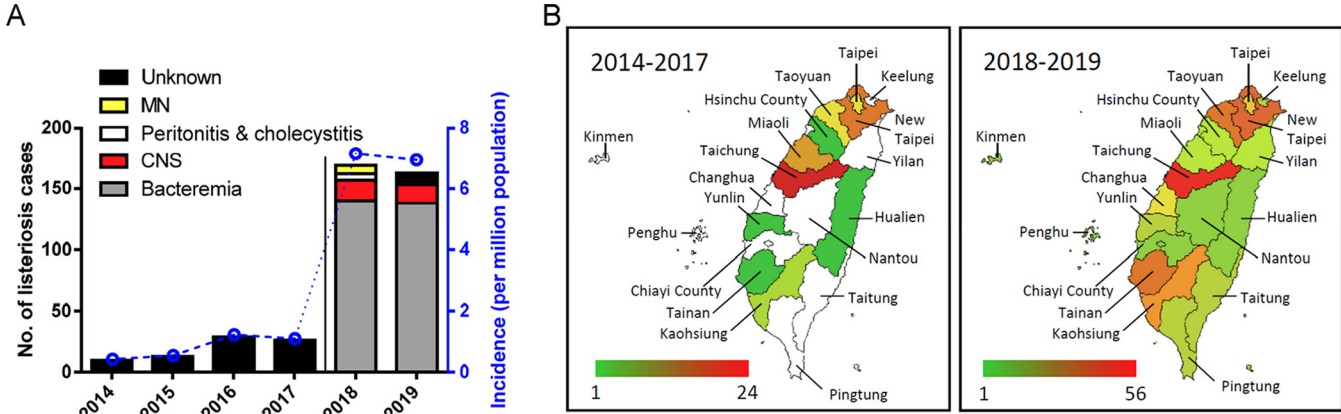

**FIG 1** Incidence of listeriosis in Taiwan. A total of nonredundant 411 clinical isolates between January 2014 and December 2019 were collected. (A) Total no. of listeriosis cases and incidence per million population before (2014 to 2017) and after (2018 to 2019) mandatory notification in Taiwan. (B) Total no. of listeriosis cases in the different administrative regions of Taiwan before (2014 to 2017) and after (2018 to 2019) mandatory. The regions in white color indicates no reported case in the period. The geographical map template was from amCharts and was used with permission.

meat sampled from open-air markets were negative for *L. monocytogenes*. In contrast, all the vegetable products sampled in 2019 (*n* = 50) were negative for *L. monocytogenes*.

**Molecular diversity of *L. monocytogenes* clinical and food isolates.** A total of 493 isolates were sequenced (Table S2). Clinical isolates (*n* = 411) were distributed across lineages I (*n* = 267, 65%) and II (*n* = 144, 35%) and 22 different cgMLST sublineages (of 20 MLST CCs). SL87 (*n* = 147, 36%), SL5 (*n* = 62, 15%) and SL378 (*n* = 58, 14%) were the most prevalent sublineages, accounting for 65% of clinical cases (Fig. S1B). SL5 was associated with CNS infection (odds ratio 2.495, *P* = 0.0373) (Fig. S1A).

**Table 1** Demographic data and *L. monocytogenes* clone distribution of the 327 human listeriosis cases between January 1, 2018 and December 31, 2019[a]

| Disease | Bacteremia | CNS infection | Peritonitis & cholecystitis | Maternal-neonatal infection | All diseases |
|---|---|---|---|---|---|
| Total | 278 | 32 | 7 | 9 | 327 |
| | | | | | |
| Clinical information | | | | | |
| Female | 145 (52%) | 17 (53%) | 3 (43%) | 7 (77.8%) | 173 (52.9%) |
| Age[b] (year) | 68.5 (0, 96) | 65.5 (20, 93) | 60 (41, 87) | 30 (26, 38)[c] | 67 (0, 96)[c] |
| Diarrhea | 18 (6.47%) | 0 (0%) | 0 (0%) | 0 (0%) | 19 (5.71%) |
| 7-day mortality (%) | 34 (12.2%) | 4 (12.5%) | 1 (14.3%) | 0[c] | 39 (11.9%)[d] |
| | | | | | |
| Lineages | | | | | |
| I | 178 (64%) | 27 (84.4%) | 4 (57.1%) | 6 (66.7%) | 215 (66%) |
| II | 100 (36%) | 5 (35.6%) | 3 (42.9%) | 3 (33.3%) | 112 (34%) |
| | | | | | |
| Serogroups | | | | | |
| IVb | 22 (7.9%) | 3 (9.4%) | 1 (14.2%) | 0 (0%) | 26 (7.95%) |
| IIb | 156 (56.1%) | 24 (75%) | 3 (42.9%) | 6 (66.7%) | 189 (57.80%) |
| IIa | 97 (34.9%) | 5 (15.6) | 3 (42.9%) | 3 (33.3%) | 109 (33.33%) |
| IIc | 3 (1.1%) | 0 (0%) | 0 (0%) | 0 (0%) | 3 (0.92%) |
| | | | | | |
| Sublineages (clonal complexes) | | | | | |
| SL87 (CC87) | 104 (37.4%) | 12 (37.5%) | 2 (28.6%) | 4 (44.5%) | 122 (37.31%) |
| SL5 (CC5) | 37 (13.3%) | 9 (28.13%) | 1 (14.3%) | 2 (22.2%) | 49 (14.98%) |
| SL378 (CC19) | 41 (14.7%) | 2 (6.25%) | 1 (14.3%) | 1 (11.1%) | 45 (13.76%) |
| SL155 (CC155) | 22 (7.9%) | 2 (6.25%) | 0 (0%) | 2 (22.2%) | 27 (8.26%) |
| SL1 (CC1) | 13 (4.7%) | 1 (3.12%) | 0 (0%) | 0 (0%) | 14 (4.28%) |
| Other SLs/CCs | 61 (22.0%) | 6 (18.75%) | 3 (42.9%) | 0 (0%) | 70 (21.41%) |

[a]Data are represented as number (percentage) unless described.
[b]Age is presented as median (min, max).
[c]The age of neonatal infection was excluded.
[d]Perinatal infections were excluded.

**Table 2** cgMLST types detected in this study comprising more than two isolates (*n* = 33 out of 117)

| cgMLST type (clonal complex, serogroup) | No. isolates (%, *N* = 493) | No. clinical isolates (%, *n* = 411) | No. food isolates (food type[b]) | Isolation years | No. regions |
|---|---|---|---|---|---|
| **Lineage I** | | | | | |
| **L1-SL87-ST87-CT58 (CC87, IIb)**[a] | **77 (15.6%)** | **54 (13%)** | **23 (13 C, 10 P)** | **2014 to 2019** | **17** |
| **L1-SL87-ST87-CT4373 (CC87, IIb)** | **53 (10.8%)** | **53 (13%)** | | **2014 to 2019** | **16** |
| **L1-SL5-ST5-CT4358 (CC5, IIb)** | **42 (8.5%)** | **42 (10%)** | | **2014 to 2019** | **9** |
| **L1-SL87-ST87-CT4374 (CC87, IIb)** | **28 (5.7%)** | **28 (7%)** | | **2016 to 2019** | **8** |
| **L1-SL1-ST1/ST1533-CT2384 (CC1, IVb)** | **11 (2.2%)** | **11 (3%)** | | **2015, 2017 to 2019** | **6** |
| L1-SL5-ST5-CT4360 (CC5, IIb) | 5 (1.0%) | 5 (1%) | | 2016, 2018 to 2019 | 3 |
| L1-SL3-ST3-CT4369 (CC3, IIb) | 4 (0.8%) | 4 (1%) | | 2016, 2018 to 2019 | 3 |
| L1-SL288-ST330-CT5502 (CC288, IIb) | 3 (0.6%) | 1 (0.2%) | 2 (P) | 2019 | 2 |
| L1-SL87-ST87-CT9095 (CC87, IIb) | 3 (0.6%) | 3 (1%) | | 2018 | 3 |
| L1-SL1-ST1-CT4353 (CC1, IVb) | 2 (0.4%) | 2 (0.5%) | | 2014 | 1 |
| L1-SL288-ST330-CT5035 (CC288, IIb) | 2 (0.4%) | 2 (0.5%) | | 2018 to 2019 | 1 |
| L1-SL5-ST5-CT4359 (CC5, IIb) | 2 (0.4%) | 2 (0.5%) | | 2015, 2019 | 1 |
| L1-SL5-ST5-CT5033 (CC5, IIb) | 2 (0.4%) | 2 (0.5%) | | 2018 | 2 |
| L1-SL5-ST5-CT5501 (CC5, IIb) | 2 (0.4%) | 2 (0.5%) | | 2019 | 1 |
| L1-SL87-ST87-CT4372 (CC87, IIb) | 2 (0.4%) | | 2 (C) | 2016 | 1 |
| L1-SL87-ST87-CT4376 (CC87, IIb) | 2 (0.4%) | 2 (0.5%) | | 2015 to 2019 | 2 |
| **Lineage II** | | | | | |
| **L2-SL378-ST378-CT4349 (CC19, IIa)** | **77 (15.6%)** | **57 (14%)** | **20 (7 C, 13 P)** | **2015 to 2019** | **12** |
| **L2-SL101-ST101-CT4370 (CC101, IIa)** | **18 (3.7%)** | **18 (4%)** | | **2016 to 2019** | **7** |
| **L2-SL155-ST155-CT4362 (CC155, IIa)** | **10 (2.0%)** | **10 (2%)** | | **2016 to 2019** | **7** |
| L2-SL9-ST9-CT4379 (CC9, IIc) | 11 (2.2%) | 1 (0.2%) | 10 (1 C, 9 P) | 2016, 2019 | 4 |
| L2-SL177-ST177-CT5041 (CC177, IIa) | 8 (1.6%) | 8 (2%) | | 2018 to 2019 | 6 |
| L2-SL155-ST155-CT4363 (CC155, IIa) | 7 (1.4%) | 7 (2%) | | 2016 to 2019 | 4 |
| L2-SL155-ST155-CT4364 (CC155, IIa) | 6 (1.2%) | 2 (0.5%) | 4 (P) | 2014 to 2015, 2019 | 4 |
| L2-SL1081-ST1081-CT4378 (ST1081, IIa) | 5 (1.0%) | 5 (1%) | | 2016, 2018 to 2019 | 3 |
| L2-SL155-ST155-CT5049 (CC155, IIa) | 5 (1.0%) | 5 (1%) | | 2018 to 2019 | 4 |
| L2-SL8-ST551-CT4347 (CC8, IIa) | 5 (1.0%) | 1 (0.2%) | 4 (C) | 2016, 2019 | 2 |
| L2-SL155-ST155-CT5050 (CC155, IIa) | 4 (0.8%) | 3 (1%) | 1 (P) | 2018 to 2019 | 4 |
| L2-SL155-ST1524-CT9039 (CC155, IIa) | 3 (0.6%) | 3 (1%) | | 2018 to 2019 | 3 |
| L2-SL155-ST155-CT4365 (CC155, IIa) | 2 (0.4%) | 2 (0.5%) | | 2015 to 2016 | 2 |
| L2-SL155-ST155-CT5496 (CC155, IIa) | 2 (0.4%) | | 2 | 2019 | 2 |
| L2-SL155-ST155-CT5539 (CC155, IIa) | 2 (0.4%) | | 2 | 2019 | 1 |
| L2-SL26-ST26-CT5754 (CC26, IIa) | 2 (0.4%) | 2 (0.5%) | | 2019 | 2 |
| L2-SL9-ST9-CT5533 (CC9, IIc) | 2 (0.4%) | | 2 | 2019 | 1 |

[a]cgMLST types with 10 or more clinical isolates are highlighted in bold.
[b]Food type: C, chicken; P, pork.

Food isolates (*n* = 82) were distributed across lineages I (*n* = 31, 38%) and II (*n* = 51, 62%) and eight different cgMLST sublineages (of seven CCs). The three most prevalent sublineages were SL87 (*n* = 26, 32%), SL378 (*n* = 21, 26%), and SL9 (*n* = 15, 18%), which accounted for 76% of the food isolates. SL87, SL378, SL155, and SL9 predominated the raw meat products (*n* = 72, 87.8%) (Fig. S1B and C). SL155 was only isolated from raw ground pork products (*n* = 10, 20.4% of the pork products) and SL8 was only isolated from raw chicken products (*n* = 4, 12.1% of the chicken products) (Fig. S1C). Although frequently isolated from listeriosis patients, neither SL5 nor SL1 was isolated from the food products (Fig. S1B).

A total of 117 cgMLST types (CT) were identified: 96 (*n* = 291 isolates) found exclusively in clinical cases, 14 exclusively in food (*n* = 18), and seven found in both human and food isolates (*n* = 184). Seventy-seven CTs (18% of clinical cases) comprised only one clinical isolate from each, and therefore, these cases were considered sporadic. All isolates collected from the same food sample belonged to the same CT, except in two raw ground pork products sampled in Taichung, in which two different strains were observed in each (cgMLST types L1-SL87-ST87-CT58 and L2-SL378-ST378-CT4349, and L1-SL3-ST3-CT5507 and L1-SL288-ST330-CT5502).

There were 26 CTs (26/103, 25%) comprising more than one clinical case and 22 spanning multiple years (Table 2), suggesting high prevalence of common CTs for several years, which may imply unnoticed long-term outbreaks. There were five cgMLST types L1-SL87-ST87-CT58 (*n* = 54, 14%), L1-SL87-ST87-CT4373 (*n* = 53, 10%), L1-SL5-ST5-CT4358

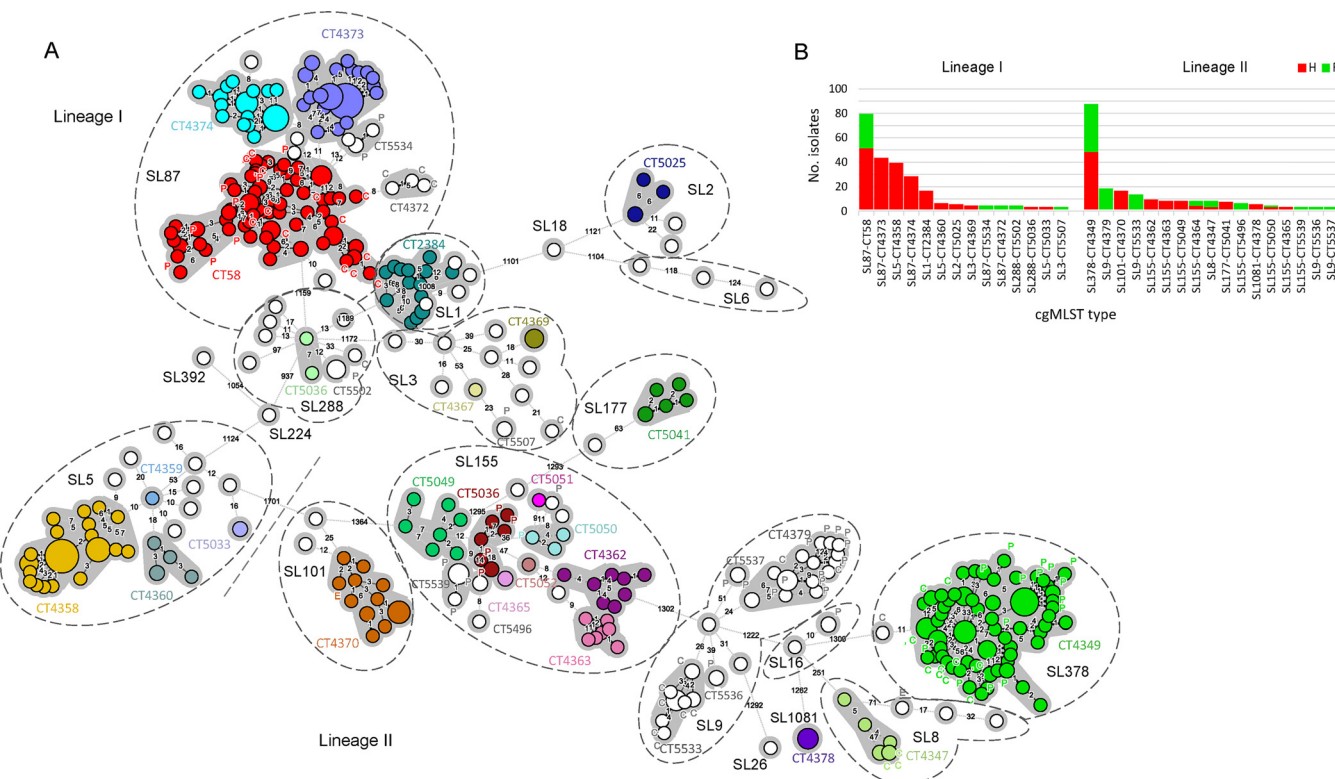

**FIG 2** cgMLST analyses of the 493 *L. monocytogenes* isolates collected between January 2014 and December 2019. (A) Minimum spanning tree of isolates' core genome MLST allelic profiles (based on 1748 loci). cgMLST profiles are represented by circles proportional to the number of isolates and the number of allelic distances between profiles are indicated in the connecting lines. The dashed bar delimits *L. monocytogenes* phylogenetic lineages, dashed circles delimit sublineages (SL; i.e., isolates sharing 150 or fewer allelic differences) and grey shadows denote clusters of isolates within the same cgMLST type (CT; i.e., isolates sharing 7 or fewer allelic differences). CT names are shown for clusters with 2 or more isolates. CTs comprising 1 or more human isolates are shown in colors. Nonclinical isolates are labeled as E (environmental), P (pork) and C (chicken). (B) Prevalence of cgMLST types with 2 or more isolates according to phylogenetic lineage and isolation source.

($n = 42$, 13%), L1-SL87-ST87-CT4374 ($n = 28$, 7%), and L2-SL378-ST378-CT4349 ($n = 57$, 13%) accounting for 57% of human listeriosis spanning up to 6 years (Table 2 and Fig. 2B). These CTs also predominated in CNS and maternal-neonatal infections in 2018 to 2019 for which disease information was available (62.5%) (Fig. S1D). While L1-SL87-ST87-CT58, L1-SL87-ST87-CT4373, and L2-SL378-ST378-CT4349 were widely spread across regions and did not exhibit a city-specific distribution, L1-SL5-ST5-CT4358 (odds ratio 8.274, $P < 0.0001$) and L1-SL87-ST87-CT4374 (odds ratio 4.084, $P = 0.0024$) were highly associated with Taichung city (Fig. S1E). In addition, 10 CTs (10/117, 8.5%) detected in this study have been previously reported in the context of listeriosis surveillance in North America, Europe, Oceania, and Asia (Table S3).

**Virulence and stress resistance genetic traits.** Relevant traits of virulence and resistance were analyzed in all clinical and food isolates. The presence of virulence traits was consistent with isolate phylogeny. All the isolates carry intact *Listeria* Pathogenicity Island 1 (LIPI-1), which contains the virulence transcription regulator *prfA,* the listeriolysin gene *hly,* and multiple genes involved in intracellular vacuole disruption (25) (Fig. 3). LIPI-3, which includes the listeriolysin S gene targeting intestinal bacteria (26, 27), was found in 11% (55/493) of the isolates in SL1, SL3, SL219, SL224, SL288, and SL455 of lineage I (Fig. 3; Table S2). LIPI-4, which contributes to *L. monocytogenes* placental and blood-CNS barrier crossing in CC4 (15), is present in 35% (175/493) of the isolates in SL87 (CC87, serogroup IIb), SL219 (CC4, serogroup IVb), and SL455 (CC455, serogroup IVb) (Fig. 3). Almost all the isolates (473/493, 96%) carry an *inlA* gene encoding for full-length internalin, a *L. monocytogenes* invasion protein involved in the crossing of the intestinal and placental barriers (14, 16, 17) (Fig. 3). In the remaining 4% isolates, mutations in *inlA* leading to premature stop codons (PMSC) were detected, resulting in the expression of a truncated InlA. InlA mutations identified concerned

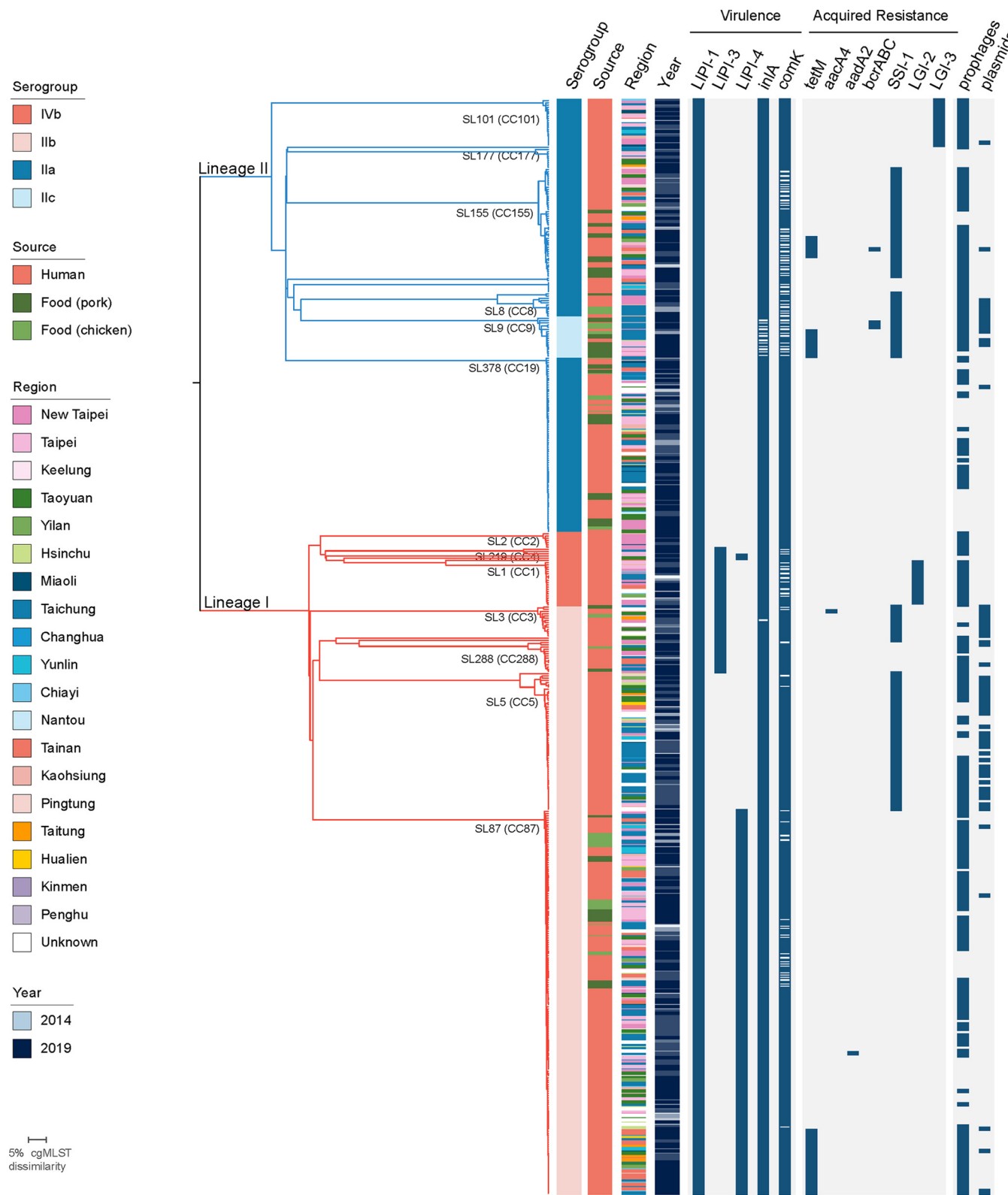

**FIG 3** Genetic diversity of the 493 *L. monocytogenes* isolates. A) Single linkage clustering based on the cgMLST profiles. Branches are colored by phylogenetic lineage (L1, red; L2, blue). Sublineages (SL) with more than 5 isolates are labelled. Information on the serogroup, year of isolation and source is provided, using the color code indicated in the legend. The presence of virulence, acquired resistance traits, prophages and plasmids are shown as blue color-filled boxes.

all SL9 isolates (previously reported PMSC_4 [*n* = 15], PMSC_8 [*n* = 1], and PMSC_11 [*n* = 2]) (28, 29), and two SL3 isolates that carry a new PMSC. The new *inlA* PMSC (termed PMSC_32, following the existing nomenclature [8, 30]) consists of a C→A substitution at nucleotide position 1041, within the leucine-rich repeat coding domain, leading to a shorter unanchored InlA protein of 346 amino acids.

Acquired resistance genes towards antibiotics were found in 52 isolates, coding for resistance towards tetracycline (*tetM*, *n* = 50; 10.1%) and aminoglycosides (*aadA2*, *n* = 1; *aacA4*, *n* = 1) (Fig. 3). Genetic traits involved in resistance towards metals, antiseptics, and/or biofilm formation were also found. These included the arsenic resistance listeria genomic islands LGI-2 (*n* = 27, in SL1 and SL2) and LGI-3 (*n* = 21, in SL101) involved in metal resistance, the stress survival islet 1 (SSI-1, *n* = 157, mainly in SL3, SL5, SL8, SL9, and SL155), and the resistance cassette *bcrABC* conferring resistance to benzalkonium chloride (*n* = 4, in SL9 and SL155) (31–34) (Fig. 3). Disrupted *comK*, proposed to be associated with biofilm formation on food-conditioned surfaces (35), was found in 29% (143/493) of the isolates (Fig. 3), and was significantly more prevalent in food isolates compared with clinical isolates ($P < 0.05$, chi square test). All these traits have been shown to confer tolerance to environmental stress in *L. monocytogenes*, facilitating its long-term persistence in production environments (32, 33, 36, 37).

## DISCUSSION

Here, we described the findings of a genomic nationwide study on clinical and food *L. monocytogenes* isolates over a 6-year period, prior to and after mandatory notification of human listeriosis cases. The incidence of listeriosis has sharply increased from 0.83 to 7 cases per million population after the implementation of mandatory notification of listeriosis in 2018, illustrating the effectiveness of mandatory notification. As a result, since July 2021, the Taiwan Food and Drug Administration has extended food monitoring of *Listeria* from infant and dairy products to refrigerated and ready-to-eat food products (https://consumer.fda.gov.tw). Compared with the regulation before this date where food products were randomly sampled once, an extended sampling plan (five to 10 sampling times per location according to the guidelines on sampling CAC GL 50-2004) is now in place for food microorganisms. This may help to identify more contamination sources of *L. monocytogenes* in different food products.

Compared with Europe and North America where pregnancy-associated listeriosis accounts for 9% to 14% of human listeriosis, only 2.8% of listeriosis cases in Taiwan were pregnancy-associated (2, 38, 39). This might be due to the difference in placental barrier crossing ability of circulating clones, and/or to the low total fertility rate in Taiwan, which is ranked the last in the world (https://www.cia.gov/the-world-factbook/field/total-fertility-rate/). Median patient age (67 years) in nonmaternal-neonatal listeriosis was slightly lower than previously reported in Europe and North America (67 to 77 years) (2, 38, 39). Here, all-cause mortality (11.9%, within 7 days) in nonmaternal-neonatal listeriosis was similar to that in Germany (13%, within 7 days) but lower than previously reported in France (30% to 46%, 90 days) and the United States (21%) (2, 38, 39), although these differences might be due to varied follow-up periods. As observed in our previous study of listeriosis cases in a medical center in Taiwan, the all-cause mortality was 14.8% within 7 days and 25.2% within 30 days (20, 39). Thus, the mortality of listeriosis is likely underestimated when considering a short follow-up period, as listeriosis can have an impact on mortality at least up to 3 months (2).

While in Western countries serogroup IVb is the main cause of human listeriosis (9, 15, 18), in Taiwan serogroups IIb (SL87/CC87, SL5/CC5, SL378/CC19) and IIa (SL155/CC155) are responsible for the majority of human listeriosis cases (74.5%, 306 of 411). Previously, SL87/CC87 (serogroup IIb) has been also reported to be responsible for 28% to 34% of clinical infections in China and is highly prevalent in raw meat products (40–42). These geographical differences may result from different food habits in Asia because serotype IVb (particularly SL1/CC1) appears to be strongly associated with dairy products (15, 18), and dairy consumption in Taiwan is relatively low across all age groups (43).

Whole-genome sequencing of isolates collected prior to and after mandatory notification allowed the identification of multiple local and nationwide clusters of cases spanning up to six years. For instance, L1-SL87-ST87-CT58 (CC87, serogroup IIb) and L2-SL378-ST378-CT4349 (CC19, serogroup IIa) comprised both clinical and food (chicken and pork) isolates distributed across multiple geographic regions from 2014 to 2019. The presence of identical cgMLST types across locations and along the years suggests long-term contamination of either food-processing plants, meat refrigeration, and/or large-scale distribution facilities, although trace-back investigations are needed to confirm the source of the outbreaks. In contrast, the region-associated distribution of L1-SL5-ST5-CT4358 to Taichung city suggests a local source of contamination. Food sampling extended to abattoir environments and meat distribution chains will help to identify the sources of contamination, if any (44). While up to 82% (96/117) of cgMLST types in this study were exclusively seen in clinical isolates, a more extensive food sampling will also help identifying the sources of contamination, which remain unknown for most clusters observed (82%, 96/117). Of note, 18% of clinical cases were due to unique cgMLST types, and are considered so far as sporadic, possible resulting from inappropriate food handling (e.g., patient's refrigerator contamination, prolonged food storage, ingestion of undercooked or unproperly washed food). The introduction of new regulation on food microorganisms sampling in 2021 will help to identify the environmental niches of *Listeria* in Taiwan.

The majority (91%) of cgMLST types found were unique to this study, reinforcing the importance of comprehensive national surveillance programs to control *Listeria* transmission (45). Nevertheless, 10 CTs have been previously reported within the scope of listeriosis surveillance in other countries (Table S3), suggestive of punctual cross-country contaminations or, in the case L1-SL87-ST87-CT58, of contamination sources with large scale distributions.

This study constitutes the first nationwide genome-based characterization of *L. monocytogenes* genotypes circulating in Taiwan, and highlights the importance of mandatory declaration in understanding listeriosis incidence. Continuous real-time genome-based surveillance coupled with epidemiological investigations and regular surveillance of food products for *L. monocytogenes* will allow to reducing contaminating sources and lower the disease burden in Taiwan.

## MATERIALS AND METHODS

**Sampling.** We collected 411 clinical *L. monocytogenes* isolates from 2014 to 2019, of which 333 were obtained in the context of the nationwide mandatory notification between January 1, 2018 and December 31, 2019 by the Taiwan Centers for Disease Control (CDC). The use of mandatory notification data were approved by the ethics committee of Institutional Review Board of Centers for Disease Control, Ministry of Health and Welfare, Taiwan. To assist epidemiological investigations, 280 food samples (113 raw chicken, 117 raw ground pork, and 50 ready-to-eat vegetable mixed salad products) (Table S1) were also collected in 2016 and 2019 from supermarkets and open-air markets in the cities where the incidence of human listeriosis was the highest (Taipei, New Taipei, and Taichung) and in Hsinchu (Table S1). Each food item of individual markets was sampled once for vegetable products from supermarkets and meat products from open-air markets. Raw meat commodities from supermarkets were sampled twice 2 weeks apart. The raw meat products were sampled from 46 markets of five districts in Taichung, 16 markets of four districts in Taipei, and 16 markets of three districts in New Taipei cities. *L. monocytogenes* isolation from food samples was performed according to the guidelines of the Taiwan food and drug administration (http://www.fda.gov.tw). Briefly, 25 g of each food sample was pre-enriched in 100 mL of *Listeria* enrichment broth UVM (CMP, Taiwan) and incubated at 37°C, 200 rpm for 18 h to 24 h. Secondary enrichment was performed by transferring 100 $\mu$L of the overnight culture to 10 mL of *Listeria* enrichment Fraser broth. After overnight incubation at 37°C with 200 rpm agitation, 10 $\mu$L of the secondary enrichment broth was seeded onto chromogenic listeria agar (CMP, Taiwan) and incubated at 37°C for up to 48 h. The suspected colonies were purified in brain heart infusion agar (CMP, Taiwan) cultured at 37°C for 18 h to 24 h. Species identification of the colonies was confirmed by MALDI-TOF mass-spectrometry as previously described (46). To reduce redundancy, isolates obtained from the same sample and belonging to the same CT were considered duplicates, and only one representative was considered in epidemiological analyses.

**Whole-genome sequencing and sequence analysis.** DNA libraries were prepared with Illumina DNA Prep and TruSeq DNA PCR-Free Low Throughput Library Preparation Kits (Illumina Inc., USA). Whole-genome sequencing was performed using Illumina MiSeq sequencing platform (Illumina Inc., USA) with MiSeq Reagent Kit v3 (2× 300 bp). Reads were trimmed with AlienTrimer (47) and assembled with SPAdes 3.11. Reads were trimmed from adapter sequences and nonconfident bases using AlienTrimmer v.0.455 (minimum read length of 30 bases and minimum quality Phred score 20, i.e., 99% base call accuracy) and corrected with Musket v.1.156, implemented in fqCleaner v.3.0 (Alexis Criscuolo, Institut Pasteur). High-accuracy assemblies were obtained from using SPAdes 3.11 with the automatic k-mer, –only-assembler and –careful options, to reduce the number of mismatches and short indels and used in downstream analyses

(48). Molecular typing based on genoserogrouping, MLST, and cgMLST was performed with BIGSdb v.1.30 (https://bigsdb.pasteur.fr/listeria/) (8, 49). The presence of plasmids, intact prophages, and *Listeria* genomic regions was inferred from the assemblies using MOB-suite v.2.0.1 (50), PHASTER (51), and BIGSdb-*Lm* (http://bigsdb.pasteur.fr/listeria/), respectively. Antibiotic resistance genes were detected using BIGSdb-*Lm* and ABRicate v.1.0.1 (https://github.com/tseemann/abricate). Minimum spanning trees and single linkage dendrograms were built from cgMLST profiles using BioNumerics v.7.6 and visualized in iTOL.

**Statistical analysis.** The significance of associations of clones with each disease type or region was analyzed with $\chi^2$ tests as described by creating a contingency table for each clone (15).

**Data availability.** All sequence data are available on NCBI Sequence Read Archive under the BioProject accession number PRJNA493675.

## SUPPLEMENTAL MATERIAL

Supplemental material is available online only.
**SUPPLEMENTAL FILE 1**, PDF file, 0.3 MB.
**SUPPLEMENTAL FILE 2**, XLSX file, 0.1 MB.

## ACKNOWLEDGMENTS

This work was supported by Ministry of Health and Welfare, Taiwan (grant numbers DOH102-DC-2102 [2013], MOHW108-CDC-C-315-112520 [2019], MOHW109-CDC-C-315-122415 [2020]), Ministry of Science and Technology, Taiwan (grant number MOST 109-2320-B-010-039), National Health Research Institutes, Taiwan (grant number NHRI-EX111-11120SC), Institut Pasteur and Inserm.

We have no conflicts to disclose.

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
