## [Reviewer comments · Microbiology Spectrum]

Microbiology Spectrum

Genomic surveillance of *Listeria monocytogenes* in Taiwan, 2014-2019

Yu-Huan Tsai, Alexandra MOURA, Zi-Qi Gu, Jui-Hsien Chang, Ying-Shu Liao, Ru-Hsiou Teng, Kuo-Yao Tseng, Dai-Ling Chang, Wei-Ren Liu, Yu-Tsung Huang, Alexandre LECLERCQ, Hsiu-Jung Lo, Marc Lecuit, and Chien-Shun Chiou

Corresponding Author(s): Chien-Shun Chiou, Centers for Disease Control

Review Timeline:

Submission Date:	May 23, 2022
Editorial Decision:	June 16, 2022
Revision Received:	July 30, 2022
Editorial Decision:	August 23, 2022
Revision Received:	September 12, 2022
Accepted:	September 20, 2022

Editor: Sandeep Tamber

Reviewer(s): The reviewers have opted to remain anonymous.

Transaction Report:

DOI: <https://doi.org/10.1128/spectrum.01825-22>

June 16, 2022

Dr. Chien-Shun Chiou
Centers for Disease Control
Centers for Diagnostics and Vaccine Development
5F 20 Wen-Sin South 3rd Road
Taichung 40855
Taiwan

Re: Spectrum01825-22 (Genomic surveillance of *Listeria monocytogenes* in Taiwan, 2014-2019)

Dear Dr. Chien-Shun Chiou:

Link Not Available

Sincerely,

Sandeep Tamber

Journals Department
Editor comments:

In addition to the reviewer comments below, please provide more details on the food sampling methodology (rationale for selection, vegetable types, was each commodity sampled every two weeks, how many markets per city, per sampling time? were the sampling locations kept constant throughout the study period?)

L96 raw ground pork, and vegetables (add the comma)

In the text can tie your findings together, for example, were any linkages or trends were noted in virulence, stress, AMR gene presence and clinical/food isolation source

The manuscript states 493 strains were sequenced. The Bioproject related to this manuscript contains 422 entries in the BioSample, and 445 in the SRA. Please correct this discrepancy.

Reviewer comments:

Reviewer #1 (Comments for the Author):

Introduction:

Line 60: delete "(Central nervous system (CNS) infection)"

Line 72: change "e.g." to "e.g.," throughout manuscript

Line 75: change "16) and showing higher" to "16). These exhibit higher..."

Methods:

Line 96: why were raw chicken, raw ground pork, and vegetables chosen? Specifically, raw chicken and raw ground pork would not be consumed raw? As well, need to be specific with respect to vegetables - which ones, condition, etc. As well, how representative is this choice of commodities?

Line 101: why 100 ml? All culture methods usually have a 1:10 start?

Line 119: what is "--only-assembler and --careful options" mean?

Results:

Line 144: "327 human listeriosis cases" - on line 483 it lists 411?

Line 165: as previous comment, please identify what vegetables were chosen and provide data (are they raw, transformed, etc.)

Line 185: "suggesting unnoticed long-term outbreaks." - is there any epidemiological evidence to support this? Otherwise, suggest to tone down the sentence as it may be a common CT may be prevalent?

Discussion:

Line 246: "of circulating clones" - how would this be tested?

Line 246: with respect to low fertility rates, is this actually the inability of women to conceive? Or that people are not having children? As well, can you not average/normalize the numbers/cases to get a better indication of the comparison to Europe and North America

Line 271: Not sure that it is suggestive of long-term contamination as there is no retail data/survey or any epidemiological evidence to support that claim. Remove and/or re-write.

General comments:

- try not to start sentences with numbers

- Table 1 (and elsewhere) - is it easier to speak about CCs (authors talk about SLs, etc. which may be less common vernacular

- Table 2 - hard to see the gray boxes? Consider bolding?

- Figure 2 - of the 493 isolates, are any duplicates from the same sample? Authors should mention this. As well, are there samples where there was actual diversity (i.e., more than one CC, etc?)

Reviewer #2 (Comments for the Author):

Authors describe their results as "a nationwide study between 2014 and 2019 of both clinical and food *Listeria monocytogenes* isolates and sequenced their genomes."

In my opinion, if this (nationwide study) is true for clinical cases of listeriosis, it is not true for food. As far as I understood, foods sampled are not high risk-foods for listeriosis. Please provide further information if I'm not correct.

Line 34: I think it was not the "observed" but the "reported"

Line 63: "0.1 and 12 per million people" Where?

Line 64: Please provide the latest reports (3, 4).

Line 65 and elsewhere in the manuscript: numbers less than ten should be written out in full

Line 71 and elsewhere in the manuscript: Avoid abbreviations at the beginning of a sentence

Lines 91-93: Is Taiwan Centers for Disease Control the only institution receiving information/ isolates from listeriosis cases in the country? Please provide information

Lines 93-95: Do you mean ethical approval for this study?

Lines 95-96: Please provide more detailed information for: "To assist epidemiological investigations, food samples (raw chicken, raw ground pork and vegetables)...". As listeriosis is mainly associated with the consumption of ready-to-eat products and is destroyed during cooking, please explain how these results could "assist epidemiological investigations"

Line 102 and elsewhere in the manuscript: insert space between "37" and "{degree sign}C"

Lines 102-106 and elsewhere in the manuscript: Please avoid long sentences. Rephrase "Then 0.1 ml of the overnight culture was transferred to 10 ml of *Listeria* enrichment Fraser broth and incubated at 37{degree sign}C, 200 rpm for 18 to 24 hr. The 10 µL of secondary enrichment broth was seeded onto Chromogenic *Listeria* Agar (CMP, Taiwan) and incubated at 37{degree sign}C for up to 48 h, and the suspected colonies were purified in Brain Heart Infusion Agar (CMP, Taiwan) cultured at 37{degree sign}C for 18 to 24 h.

Lines 134-135: Only found 445 sequences. Please check.

165-166: "By contrast, all the vegetable products sampled in 2019 (n=50) were negative for *L. monocytogenes*. " What type of products?

Lines 181: Are the sample products eaten raw?

Lines 185: or widely distributed CTs?

Line 231: in my opinion, we can not claim "epidemiological ... study"

Lines 235-242- "As a result, since July 2021, the Taiwan Food and Drug Administration (TFDA) has extended food monitoring of *Listeria* from infant and dairy products to refrigerated and ready-to-eat food products (<https://consumer.fda.gov.tw>). As compared to the regulation before this date where food products were randomly sampled once, an extended sampling plan (5 to 10 sampling times per location according to the guidelines on sampling CAC GL 50-2004) is now in place for food microorganisms. This may help to identify more contamination sources of *L. monocytogenes* in different food products." Useful information but probably not in this context"

Lines 243-245 "As compared to Europe and North America" this should be supported with references from official entities.

Fig 3- The figure looks very nice, but it is difficult to read. A supplementary table with all the information for each isolate would be useful.

Staff Comments:

Preparing Revision Guidelines

Please return the manuscript within 60 days; if you cannot complete the modification within this time period, please contact me. If you do not wish to modify the manuscript and prefer to submit it to another journal, please notify me of your decision immediately so that the manuscript may be formally withdrawn from consideration by Microbiology Spectrum.

We appreciated the valuable comments from editor and reviewers. We answered the questions/concerns in the following text, made several modifications and rephrased paragraph especially on the details of food sampling. We provided a table describing food sampling details (Table S1), and a table that links to Figure 3, summarizing genomic characteristics of the 493 isolates included in this study (Table S2). We hope we have addressed all the points and sincerely thank for your interest of our work.

Editor comments:

In addition to the reviewer comments below, please provide more details on the food sampling methodology (rationale for selection, vegetable types, was each commodity sampled every two weeks, how many markets per city, per sampling time? were the sampling locations kept constant throughout the study period?)

We focused on chicken and pork products as these food commodities are the two main meat types consumed by Taiwanese, and are mostly produced locally in Taiwan. Foods made by these meat types are at high risk of being undercooked in some Chinese cuisine styles, thus a likely source of contamination. We included now the detailed sampling information in the revised manuscript (L93-101 and the new Table S1). Thank you.

L96 raw ground pork, and vegetables (add the comma)

We followed this suggestion (L94). Thank you.

In the text can tie your findings together, for example, were any linkages or trends were noted in virulence, stress, AMR gene presence and clinical/food isolation source.

Similar to other studies, virulence and resistance traits found in this study were more linked to the phylogeny of the isolate rather than the source type. Only *comK* truncations were significantly associated with food isolation source. This information is now detailed in the revised manuscript (L233-L235). Thank you.

The manuscript states 493 strains were sequenced. The Bioproject related to this manuscript contains 422 entries in the BioSample, and 445 in the SRA. Please correct this discrepancy.

The 445 SRA are those open for public in the Bioproject, and not all the sequences included in this study are publicly available now. Nevertheless, all sequences concerning the 493 isolates in this study have been deposited in SRA (Bioproject PRJNA493675) and will be made publicly available upon acceptance. Isolate accession

numbers are provided in the new Table S2).

Reviewer comments:

Reviewer #1 (Comments for the Author):

Introduction:

Line 60: delete "(Central nervous system (CNS) infection)"

We followed the suggestion and deleted this in the revised manuscript (L60).

Line 72: change "e.g." to "e.g.," throughout manuscript

Changed as suggested (L71-72).

Line 75: change "16) and showing higher" to "16). These exhibit higher..."

Changed as suggested (L74).

Methods:

Line 96: why were raw chicken, raw ground pork, and vegetables chosen? Specifically, raw chicken and raw ground pork would not be consumed raw? As well, need to be specific with respect to vegetables - which ones, condition, etc. As well, how representative is this choice of commodities?

We focused on chicken and pork products as these food commodities are the two main meat types consumed by Taiwanese, and are mostly produced locally in Taiwan. We began with raw meat products as in some Chinese cuisine styles these two meat products may be undercooked and possibly leading to cross contamination to cooked food during preparation. The 50 vegetable products are ready-to-eat mixed salad products sold in supermarkets. We aimed to have the samples representing the whole city and thus sampled 46 markets of 5 districts in Taichung, 16 markets of 4 districts in Taipei, and 16 markets of 3 districts in New Taipei cities. We included the sampling information in the revised manuscript (L93-101).

Line 101: why 100 ml? All culture methods usually have a 1:10 start?

We followed the guidance of the Taiwan FDA. As the starting food samples are not in liquid, and 25 g of meat is not easy to be homogenized. Incubation with 100 ml broth would allow complete soaking of the food samples according to our experience.

Line 119: what is "--only-assembler and --careful options" mean?

These options refer to the assembler parameters for high-accuracy assembly. This is

now clarified in the text (L123-125).

Results:

Line 144: "327 human listeriosis cases" - on line 483 it lists 411?

We have 411 human isolates in this study, of which 78 isolates were collected in 2014-2017, and 333 isolates were collected in 2018-2019 upon mandatory notification. Among the 333 isolates in 2018-2019, disease information was available from 327 human listeriosis cases. This is now made clear in the text (L143-148).

Line 165: as previous comment, please identify what vegetables were chosen and provide data (are they raw, transformed, etc.)

The vegetables are ready-to-eat mixed salad products sold in supermarkets. We included this information in the revised manuscript (L94-95 and the new Table S1).

Line 185: "suggesting unnoticed long-term outbreaks." - is there any epidemiological evidence to support this? Otherwise, suggest to tone down the sentence as it may be a common CT may be prevalent?

Several studies have shown that isolates belonging to the same cgMLST type (CT) are likely to share an epidemiological link (eg., Moura et al., *Nat. Microbiol.*, 2016; Schjørring et al., *Euro Surveill.*, 2017; Van Walle et al., *Euro Surveill.*, 2018). However, in this study we do not have trace-back trace-forward investigations to confirm the source of the outbreaks. We thus follow the suggestion and rephrased this to "suggesting high prevalence of common CTs for several years, which may imply unnoticed long-term outbreaks." (L191-193).

Discussion:

Line 246: "of circulating clones" - how would this be tested?

Previous studies have shown differences in placental barrier crossing ability of different clones (Maury, Tsai et al., *Nat Genet.*, 2016), using a humanized mouse model for listeriosis developed in 2008 (Disson et al., *Nature*, 2008). While this is known for the most prevalent clones in Western countries, the virulence and ability to cross host barriers for the most prevalent clones identified in this study (SL87/CC87 and SL378/CC19) remains to be elucidated.

Line 246: with respect to low fertility rates, is this actually the inability of women to conceive? Or that people are not having children? As well, can you not average/normalize the numbers/cases to get a better indication of the comparison to Europe and North America

According to the public information from the Central Intelligence Agency of the United States of the America, the total fertility rate is 1.84 children born/woman in the U.S.A. and 2.03 children born/woman in France, while it is 1.08 born/woman in Taiwan. However, it is difficult to do normalization with statistical power to test the difference as we may need additional information, such as the detail of total fertility rate calculation that we do not have for analysis. We also agree that the low fertility rate in Taiwan could be due to the inability of women to conceive or that people are not having children, both are correlated with each other.

Line 271: Not sure that it is suggestive of long-term contamination as there is no retail data/survey or any epidemiological evidence to support that claim. Remove and/or re-write.

We agree that analysis of isolates from food processing and retailing environments is required to make the conclusion. We thus rewrote the sentence to be clearer (L277-280).

General comments:

- try not to start sentences with numbers

We followed the comment and revised the manuscript accordingly. Thank you.

- Table 1 (and elsewhere) - is it easier to speak about CCs (authors talk about SLs, etc. which may be less common vernacular

In this study we employed cgMLST, which provides a high resolution of circulating strains in comparison to MLST. It is thus more precise to refer to SLs, which are assigned by cgMLST analysis, instead of CCs that are based on 7-gene MLST. To address the reviewer's comment and to be clearer for readers, we now include both nomenclatures in Tables and throughout the text when relevant.

- Table 2 - hard to see the gray boxes? Consider bolding?

We now highlight the largest CTs in bold to be clearer to readers. Thank you.

- Figure 2 - of the 493 isolates, are any duplicates from the same sample? Authors should mention this. As well, are there samples where there was actual diversity (i.e., more than one CC, etc?)

Multiple isolates belonging to the same CT from one sample were seen as duplicates, and we only took one from the duplicates for epidemiological analyses. We included this information in the revised manuscript (L111-113).

We have observed strain diversity in two raw ground pork products sampled in Taichung. This information is now also included in the text (L188-190).

Reviewer #2 (Comments for the Author):

Authors describe their results as "a nationwide study between 2014 and 2019 of both clinical and food *Listeria monocytogenes* isolates and sequenced their genomes."

In my opinion, if this (nationwide study) is true for clinical cases of listeriosis, it is not true for food. As far as I understood, foods sampled are not high risk-foods for listeriosis. Please provide further information if I'm not correct.

While ready-to-eat food products are the main high-risk foods for listeriosis outbreak in the literature, we could not exclude the involvement of undercooked foods, which are the main high-risk foods for infection of *Campylobacter* and *Salmonella*, in sporadic listeriosis. No cluster infection of *L. monocytogenes* at a specific place in a short period has been reported in Taiwan. We thus speculated that inappropriate food processing, such as ingestion of undercooked meat, may account for the sporadic listeriosis cases. Indeed, *L. monocytogenes* can be isolated from raw meats (Luo et al., Food Control, 2017; Wang et al., Front Med., 2018). We thus decided to begin with sampling of raw meat products in Taiwan.

Line 34: I think it was not the "observed" but the "reported"

The incidence shown here has not been reported and is one of the findings in this study. We therefore used "observed" here (L34).

Line 63: "0.1 and 12 per million people" Where?

These incidences were reported in Europe and the U.S.A.. We updated the references to those published in two years. Please see the references 3-5 in the revised manuscript.

Line 64: Please provide the latest reports (3, 4).

We followed this suggestion and updated the references to those published in two years. Please see the references 3-5 in the revised manuscript.

Line 65 and elsewhere in the manuscript: numbers less than ten should be written out in full

We followed the suggestion and revised the writing of the numbers.

Line 71 and elsewhere in the manuscript: Avoid abbreviations at the beginning of a sentence

We followed the suggestion and revised the sentences.

Lines 91-93: Is Taiwan Centers for Disease Control the only institution receiving information/ isolates from listeriosis cases in the country? Please provide information
Yes, Taiwan Centers for Disease Control is the only institution collecting the isolates and patient information of notifiable diseases, including listeriosis since 2018, in a nationwide manner in Taiwan (<https://www.cdc.gov.tw/En>). We made this clear in the text (L78-80).

Lines 93-95: Do you mean ethical approval for this study?

Yes. We provided further explanation to be clearer:

“The use of mandatory notification data was approved by the ethics committee of Institutional Review Board of Centers for Disease Control, Ministry of Health and Welfare, Taiwan.” (L91-93).

Lines 95-96: Please provide more detailed information for: "To assist epidemiological investigations, food samples (raw chicken, raw ground pork and vegetables)...". As listeriosis is mainly associated with the consumption of ready-to-eat products and is destroyed during cooking, please explain how these results could "assist epidemiological investigations"

Raw and undercooked meat and poultry are one of the potential source of bacterial foodborne diseases. Processing and/or preserving the raw meat food products together with cooked or ready-to-eat food products may occur at home, or in some cases in restaurants. Moreover, in some Chinese cuisine styles the raw food products are at high risk of being undercooked. Our findings thus highlight the potential of raw and undercooked meat underlying listeriosis in Taiwan.

Line 102 and elsewhere in the manuscript: insert space between "37" and "{degree sign}C"

We followed this suggestion and revised them in the manuscript.

Lines 102-106 and elsewhere in the manuscript: Please avoid long sentences. Rephrase "Then 0.1 ml of the overnight culture was transferred to 10 ml of Listeria enrichment Fraser broth and incubated at 37{degree sign}C, 200 rpm for 18 to 24 hr. The 10 μL of secondary enrichment broth was seeded onto Chromogenic Listeria Agar (CMP, Taiwan) and incubated at 37{degree sign}C for up to 48 h, and the suspected colonies were purified in Brain Heart Infusion Agar (CMP, Taiwan) cultured at 37{degree sign}C for 18 to 24 h.

We rephrased long sentences throughout the manuscript. This part in the method was also revised (L105-110).

Lines 134-135: Only found 445 sequences. Please check.

The 445 in SRA are those open for public in the Bioproject, and not all the sequences included in this study are publicly available now. Nevertheless, all the 493 SRA sequences included in this study have been uploaded to Bioproject PRJNA493675 and will be made public upon acceptance. The accession numbers of the isolates are provided in the new Table S2.

165-166: "By contrast, all the vegetable products sampled in 2019 (n=50) were negative for *L. monocytogenes*. " What type of products?

These are ready-to-eat mixed salad products sold in supermarkets. We added this information in the revised manuscript (L93-97).

Lines 181: Are the sample products eaten raw?

People in Taiwan do not directly eat these raw meat products. However, processing and/or preserving the raw meat food products together with cooked or ready-to-eat food products may occur at home, or in some cases in restaurants. Moreover, there is a risk that the raw meat products sampled in this study may be undercooked in some Chinese cuisine styles.

Lines 185: or widely distributed CTs?

We agree that these CTs might be widely distributed as we have not investigated the potential contamination of the CTs in food processing and retailing environments as compared to those in soil and water in wild environments. We thus revised the sentence to "suggesting high prevalence of common CTs for several years, which may imply unnoticed long-term outbreaks." (L191-193).

Line 231: in my opinion, we can not claim "epidemiological ... study"

We agree that we only have preliminary findings in epidemiology and more complete in genomic surveillance. We followed this comment and revised the sentence to: "Here we described the findings of a genomic nationwide study on clinical and food *L. monocytogenes* isolates over a 6-year period, prior to and after mandatory notification of human listeriosis cases." (L240-242).

Lines 235-242- "As a result, since July 2021, the Taiwan Food and Drug Administration (TFDA) has extended food monitoring of *Listeria* from infant and dairy products to refrigerated and ready-to-eat food products (<https://consumer.fda.gov.tw>). As compared to the regulation before this date where food products were randomly sampled once, an

extended sampling plan (5 to 10 sampling times per location according to the guidelines on sampling CAC GL 50-2004) is now in place for food microorganisms. This may help to identify more contamination sources of *L. monocytogenes* in different food products." Useful information but probably not in this context"

Here we discussed how our preliminary findings changed the food sampling policy of the TFDA. The change of the policy may impact *L. monocytogenes* isolation rates in near future. Nevertheless, we can remove this part if the Editor and Reviewers insist that this is not in the context of the study.

Lines 243-245 "As compared to Europe and North America" this should be supported with references from official entities.

The references cited are from official institutions of listeriosis surveillance in U.S.A., Germany and France.

Fig 3- The figure looks very nice, but it is difficult to read. A supplementary table with all the information for each isolate would be useful.

We provide now all isolates' detailed information in a supplementary table (Table S2). Thank you.

August 23, 2022

Dr. Chien-Shun Chiou
Centers for Disease Control
Centers for Diagnostics and Vaccine Development
5F 20 Wen-Sin South 3rd Road
Taichung 40855
Taiwan

Re: Spectrum01825-22R1 (Genomic surveillance of *Listeria monocytogenes* in Taiwan, 2014-2019)

Dear Dr. Chien-Shun Chiou:

Thank you for submitting your manuscript to Microbiology Spectrum. As you will see your paper is very close to acceptance. Please modify the manuscript along the lines I have recommended. As these revisions are quite minor, I expect that you should be able to turn in the revised paper in less than 30 days, if not sooner. If your manuscript was reviewed, you will find the reviewers' comments below.

When submitting the revised version of your paper, please provide (1) point-by-point responses to the issues raised by the reviewers as file type "Response to Reviewers," not in your cover letter, and (2) a PDF file that indicates the changes from the original submission (by highlighting or underlining the changes) as file type "Marked Up Manuscript - For Review Only". Please use this link to submit your revised manuscript. Detailed instructions on submitting your revised paper are below.

Link Not Available

Sincerely,

Sandeep Tamber

Editor comments:

The objective needs to be stronger and there needs to be an emphasis that infections being investigated were likely sporadic. In response to reviewer 2 you state: "No cluster infection of *L. monocytogenes* at a specific place in a short period has been reported in Taiwan. We thus speculated that inappropriate food processing, such as ingestion of undercooked meat, may account for the sporadic listeriosis cases." This is a strong rationale, can this statement or a version of it be incorporated into the text?

Preparing Revision Guidelines

- Point-by-point responses to the issues raised by the reviewers in a file named "Response to Reviewers," NOT IN YOUR COVER LETTER.
- Upload a compare copy of the manuscript (without figures) as a "Marked-Up Manuscript" file.
- Each figure must be uploaded as a separate file, and any multipanel figures must be assembled into one file.

- Manuscript: A .DOC version of the revised manuscript
- Figures: Editable, high-resolution, individual figure files are required at revision, TIFF or EPS files are preferred

Please return the manuscript within 60 days; if you cannot complete the modification within this time period, please contact me. If you do not wish to modify the manuscript and prefer to submit it to another journal, please notify me of your decision immediately so that the manuscript may be formally withdrawn from consideration by Microbiology Spectrum.

Editor comments:

The objective needs to be stronger and there needs to be an emphasis that infections being investigated were likely sporadic. In response to reviewer 2 you state: "No cluster infection of *L. monocytogenes* at a specific place in a short period has been reported in Taiwan. We thus speculated that inappropriate food processing, such as ingestion of undercooked meat, may account for the sporadic listeriosis cases." This is a strong rationale, can this statement or a version of it be incorporated into the text?

We speculated that sporadic cases might be due to undercooked meat before this study. However, our sequence analyses of the food and clinical isolates identified the major CTs among the isolates, suggesting the potential of clustered listeriosis cases. We also identified 77 CTs that comprised only 1 isolates within each CT in clinical isolates. We consider these as sporadic cases. We include the information in both results (L187-L188) and discussion (L289-L292). Please see the marked up manuscript for the detail.

September 20, 2022

Dr. Chien-Shun Chiou
Centers for Disease Control
Centers for Diagnostics and Vaccine Development
5F 20 Wen-Sin South 3rd Road
Taichung 40855
Taiwan

Re: Spectrum01825-22R2 (Genomic surveillance of *Listeria monocytogenes* in Taiwan, 2014-2019)

Dear Dr. Chien-Shun Chiou:

Your manuscript has been accepted, and I am forwarding it to the ASM Journals Department for publication. You will be notified when your proofs are ready to be viewed.

Sincerely,

Sandeep Tamber
Editor, Microbiology Spectrum

Journals Department
Supplemental Material: Accept
Supplemental Material: Accept